# Global Prevalence of Colistin Resistance in *Klebsiella pneumoniae* from Bloodstream Infection: A Systematic Review and Meta-Analysis

**DOI:** 10.3390/pathogens11101092

**Published:** 2022-09-24

**Authors:** Leonard Ighodalo Uzairue, Ali A. Rabaan, Fumilayo Ajoke Adewumi, Obiageli Jovita Okolie, Jamiu Bello Folorunso, Muhammed A. Bakhrebah, Mohammed Garout, Wadha A. Alfouzan, Muhammad A. Halwani, Aref A. Alamri, Shaima A. Halawani, Fatimah S. Alshahrani, Abdulkarim Hasan, Abbas Al Mutair, Saad Alhumaid, Johnson Etafo, Idorenyin Utip, Ikenna Maximillian Odoh, Nkolika S. Uwaezuoke

**Affiliations:** 1Department of Medical Laboratory Science, Faculty of Basic Medical Science, Federal University, Oye-Ekiti 371104, Ekiti State, Nigeria; 2Department of Microbiology, College of Bioscience, Federal University of Agriculture, Abeokuta 111101, Ogun State, Nigeria; 3Molecular Diagnostic Laboratory, Johns Hopkins Aramco Healthcare, Dhahran 31311, Saudi Arabia; 4College of Medicine, Alfaisal University, Riyadh 11533, Saudi Arabia; 5Department of Public Health and Nutrition, The University of Haripur, Haripur 22610, Pakistan; 6Department of Medical Microbiology, College of Medicine, Ekiti State University, Ado-Ekiti 362103, Ekiti State, Nigeria; 7Department of Applied Sciences, University of West England, Bristol BS16 1QY, UK; 8Medical Microbiology Unit, Department of Medical Laboratory, Directorate of Health Services, Olabisi Onibanjo University, Ago-Iwoye 120107, Ogun State, Nigeria; 9Life Science and Environment Research Institute, King Abdulaziz City for Science and Technology (KACST), Riyadh 11442, Saudi Arabia; 10Department of Community Medicine and Health Care for Pilgrims, Faculty of Medicine, Umm Al-Qura University, Makkah 21955, Saudi Arabia; 11Department of Microbiology, Faculty of Medicine, Kuwait University, Safat 13110, Kuwait; 12Microbiology Unit, Department of Laboratories, Farwania Hospital, Farwania 85000, Kuwait; 13Department of Medical Microbiology, Faculty of Medicine, Al Baha University, Al Baha 4781, Saudi Arabia; 14Department of Molecular Microbiology and Cytogenetics, Riyadh Regional Laboratory, Riyadh 11425, Saudi Arabia; 15Division of Infectious Diseases, Department of Internal Medicine, College of Medicine, King Saud University and King Saud University Medical City, Riyadh 11451, Saudi Arabia; 16Department of Pathology, Faculty of Medicine, Al-Azhar University, Cairo 11884, Egypt; 17Prince Mishari Bin Saud Hospital in Baljurashi, Ministry of Health, Baljurash 22888, Saudi Arabia; 18Research Center, Almoosa Specialist Hospital, Al-Ahsa 36342, Saudi Arabia; 19College of Nursing, Princess Norah Bint Abdulrahman University, Riyadh 11564, Saudi Arabia; 20School of Nursing, Wollongong University, Wollongong, NSW 2522, Australia; 21Nursing Department, Prince Sultan Military College of Health Sciences, Dhahran 33048, Saudi Arabia; 22Administration of Pharmaceutical Care, Al-Ahsa Health Cluster, Ministry of Health, Al-Ahsa 31982, Saudi Arabia; 23Department of Medical Microbiology, Federal Medical Centre, Owo 341105, Ondo State, Nigeria; 24School of Medical and Health Sciences, Bangor University, Bangor LL57 2EF, UK; 25University Medical Centre, Federal University, Oye-Ekiti 371104, Ekiti State, Nigeria; 26Department of Medical Microbiology, Federal Medical Centre, Abuja 310001, Federal Capital Territory, Nigeria

**Keywords:** colistin-resistance, *Klebsiella pneumoniae*, bloodstream infection, multi-drug resistance, prevalence

## Abstract

**Background:** Among gram-negative bacteria, *Klebsiella pneumoniae* is one of the most common causes of healthcare-related infection. Bloodstream infections (BSIs) caused by *Klebsiella pneumoniae* are notorious for being difficult to treat due to resistance to commonly used antimicrobials. *Klebsiella pneumoniae* isolates from bloodstream infections are becoming increasingly resistant to carbapenems. In the fight against carbapenem-resistant *Klebsiella pneumoniae*, colistin [polymyxin E] is the antimicrobial of choice and is thus widely used. **Objective:** This study aimed to determine the global prevalence of colistin resistance amongst *Klebsiella pneumoniae* isolates from bloodstream infections. **Methods:** PubMed, Medline, Scopus, and the Cochrane Library were searched for published articles without restricting the search period. Studies meeting the predefined inclusion and exclusion criteria were included, and quality was assessed using Joanna Briggs Institute Checklist. We used a statistical random effect model to analyze data with substantial heterogeneity (I^2^ > 50%) in the meta-analysis. **Results:** A total of 10 studies out of 2873 search results that met the inclusion criteria were included in the final synthesis for this study. A pooled prevalence of colistin resistance was 3.1%, 95% CI (1.5–4.7%). The highest colistin resistance pooled prevalence was recorded in isolates studied in 2020 and beyond 12.90% (4/31), while *Klebsiella pneumoniae* isolates studied in 2015 and before and in 2016–2019 showed a pooled colistin resistance rate of 2.89% (48/1661) and 2.95% (28/948), respectively. The highest colistin resistance was found in *Klebsiella pneumoniae* isolates from Thailand (19.2%), while the least pooled resistance was in *Klebsiella pneumoniae* from South Korea (0.8%). The pooled prevalence of the multidrug-resistant (MDR) of *Klebsiella pneumoniae* from bloodstream infection ranged from 80.1%, 95% CI (65.0–95.2%), and the resistance prevalence of other antibiotics by *Klebsiella pneumoniae* from bloodstream infections were as follows; ciprofloxacin (45.3%), ertapenem (44.4%), meropenem (36.1%), imipenem (35.2%), gentamicin (33.3%), amikacin (25.4%) and tigecycline (5.1%). *Klebsiella pneumoniae* recovered from the intensive care unit (ICU) showed higher colistin resistance, 11.5% (9/781%), while non-ICU patients showed 3.03% (80/2604) pooled colistin resistance. **Conclusion:** This study showed low colistin resistance in *Klebsiella pneumoniae* isolates from global bloodstream infections. However, significant colistin resistance was observed in isolates collected from 2020 and beyond. Significant colistin resistance was also observed in *Klebsiella pneumoniae* isolates in bloodstream infections from the intensive care unit (ICU) compared to those from non-ICUs. As a result, there is a need to institute colistin administration stewardship in the ICU in clinical settings.

## 1. Introduction

Multi-drug resistant (MDR) gram-negative bacteria are becoming more widespread around the globe, which presents an increasing risk to human health [1]. Nosocomial *Klebsiella pneumoniae* is responsible for one-third of all gram-negative infections and is associated with carbapenem resistance worldwide [2,3]. These infections include urinary tract infections (UTIs), pneumonia, meningitis, and bloodstream infections. Extended-resistant *Klebsiella pneumoniae* isolates are resistant to carbapenems, fluoroquinolones, aminoglycosides, and cephalosporins. Studies by Dixit et al. [4] and Gandra et al. [5] demonstrated an increase in carbapenem-resistant *Klebsiella pneumoniae* isolates from 29% in 2008 to 57% in 2016. Azam et al. [6] stated that an increase in the carbapenem-resistant *Klebsiella pneumoniae* decreased the number of treatment choices available for potentially fatal infections. A study in Italy in 2013 showed a 43% colistin resistance from carbapenem-resistant isolates collected from 21 hospitals in the study area. It was therefore asserted by the author that there is a possible evolution of colistin resistance in a setting with high *Klebsiella pneumoniae* carbapenemase-producing *Klebsiella pneumoniae* [7].

Bloodstream infections caused by colistin-resistant (ColR) *Klebsiella pneumoniae* (Kp) strains are a cause for concern due to the potential for complications, poor clinical outcomes, and limited treatment options [8,9]. The Polymyxins (colistin and polymyxin B) have made a comeback as a treatment option for infections caused by carbapenem-resistant *Klebsiella pneumoniae* since there are very few new antimicrobials in the pipeline [10,11]. Colistin-resistant *Klebsiella pneumoniae* has a high degree of genetic plasticity; polymyxin resistance may be conferred by a point mutation and a genetic disruption in two-component regulatory systems (TCRS), including pmrAB and phoPQ [12,13].

Polymyxins are the most widely used antimicrobials in the fight against carbapenem-resistant *Klebsiella pneumoniae* [11]. The recent discovery of colistin-resistant *Klebsiella pneumoniae* isolates is a cause for worry, particularly in light of the limited antibiotic options available and the high death rate associated with these infections [14]. Thus, clinical microbiology laboratories need to identify colistin resistance to adequately respond and prevent patients from receiving ineffective and potentially nephrotoxic medicine [15]. In the past, most clinical microbiology laboratories utilized disk diffusion to evaluate antibiotics, and recently, there has been a transition from manual antibiotic susceptibility testing (AST) to automated techniques [16]. The disk diffusion method has been found to be inefficient and unreliable for colistin susceptibility testing [17]. This assertion is due to the absence of an established breakpoint for disk diffusion for colistin susceptibility testing. The CLSI and EUCAST advised broth micro-dilution to assess colistin susceptibility due to established breakpoint. However, the breakpoints for most automated AST systems for colistin susceptibility have not been established, hence the disapproval by CLSI and EUCAST on the using automated method for colistin susceptibility [18]. 

As a result of the findings above, there is a need to determine the prevalence of colistin resistance by *Klebsiella pneumoniae* from bloodstream infections. We therefore conducted a comprehensive review and meta-analysis to assess the prevalence of colistin resistance in *Klebsiella pneumoniae* isolates and their antibiotic resistance patterns.

## 2. Methods

### 2.1. Search Strategies and Database Used

The databases PubMed, Medline, Scopus, and the Cochrane Library were searched using the MeSH keywords “*Klebsiella pneumoniae*”, or *Klebsiella* and “Colistin Resistant*”, or “Polymyxin E”, or “drug-resistant*”, or “antimicrobial-resistant*” and “bloodstream infection”, or bacteremia*, or sepsis. There were no date restrictions placed on the searches. Combinations of MeSH words with “*Klebsiella pneumoniae*”, “colistin resistance”, and “antibiotic(s)” were performed. We followed the PRISMA guidelines in the study.

### 2.2. Eligibility

#### 2.2.1. Inclusion Criteria

L.I.U., F.A.A., and A.H. evaluated the titles and abstracts of the papers separately to choose the full-text articles. Another author (A.A.R.) helped resolve any disputes between the three authors. Any study reporting colistin-resistant *Klebsiella pneumoniae* from bloodstream infections that used standard laboratory methods was included in any original studies published and written in English. Studies were considered eligible if they were primary (original) studies with hospital patients with bloodstream infections and had bacterial isolation reports of *Klebsiella pneumoniae* and colistin resistance and antibiotic susceptibility testing based on criteria established by CLSI and EUCAST for colistin resistance testing used. The method used for colistin susceptibility testing was microdilution, the only recommended method for antibiotic susceptibility testing for colistin [18,19]. 

#### 2.2.2. Exclusion Criteria

The following studies were excluded: any study with a sample size of *Klebsiella pneumoniae* isolates less than 10 (examples are case reports or case series with less than ten isolates), or secondary analysis of *Klebsiella pneumoniae* isolates already reported. Studies that only showed antibiotic resistance without reporting colistin resistance were excluded from the analysis, or studies that reported colistin susceptibility testing methods other than micro-dilution. 

### 2.3. Data Extraction and Data Collection

Name of the first author, date of publication, number of participants, length of time and location of the study, *Klebsiella pneumoniae* strain identified, and resistance to colistin and other antibiotics were used. Data were collected by three writers (L.I.U., F.A.A., and A.H.), while a fourth writer (A.A.R.) reviewed the data’s validity. Figure 1 shows the flow chart of the literature screening process.

### 2.4. Quality Assessment

The revised Critical Appraisal Checklist developed by the Joanna Briggs Institute [20,21] in which disagreements are resolved by consensus. The checklist for each study required the reviewers to complete seven items, with question four having two possible answers. One point was awarded for each “Yes”. The final score for each research might fall between 0 and 8 (Appendix A).

### 2.5. Meta-Analysis Approach

Every statistical analysis was conducted using OpenMeta (analyst) Software (Brown University, Providence, RI, USA). The Chi-Square test, often known as Cochran’s Q, was performed to establish whether or not the studies were heterogeneous before pooling the data. We used a random-effect model by the DelSimonian-Laird (DL method to analyze data with substantial amounts of heterogeneity (I^2^ > 50%) [22]. The test developed by Begg was used to analyze publication bias. We determined the prevalence and 95% confidence interval by Jackson method of DelSimonian-Laird (DL) model for colistin resistance across counties and based on the source of the sample.

## 3. Results

### 3.1. Studies Selection

A total of 2873 articles were recovered from the initial search. Ten (10) studies [5,23,24,25,26,27,28,29,30,31,32] out of 2873 recovered met the predefined inclusion criteria and were included in the final analysis. Articles based on a review or case report were not considered. Moreover, studies that used methods other than micro-dilution for antibiotic susceptibility for colistin were excluded as micro-dilution is the only standard method with established breakpoints. Studies reporting different antibiotic resistance prevalence other than colistin with other classes of antibiotics were excluded from the study. The selection process and characteristics of each of the studies, including quality score, are shown in Figure 1 and Appendix A. The included studies were from seven (7) countries across four WHO regions: America, Eastern Mediterranean, South-East Asia, and Western Pacific, respectively. All of the included studies examined colistin resistance in *Klebsiella pneumoniae* isolates recovered from bloodstream infections.

### 3.2. Overall Pooled Colistin Resistance in Klebsiella pneumoniae from Bloodstream Infection in the Study 

A total of 89 *Klebsiella pneumoniae* had colistin resistance from 2856 *Klebsiella pneumoniae* isolates from bloodstream infections from the ten (10) included studies. The pooled prevalence of colistin resistance was 3.1%, 95% CI (1.5–4.7%) with a heterogeneity value (I^2^ of 84%, as shown in Figure 2. The funnel plot (Figure 3) with Egger’s regression asymmetry test (*p* < 0.05) and Kendall’s Tau test (*p* = 0.382). Figure 4 showed periodic pooled prevalence resistance of bacteremia *Klebsiella pneumoniae* from bloodstream infections, and a pooled colistin resistance of 2.89% (48/1661) was observed in isolates recovered in the year 2015 or before, a 2.95% (28/948) pooled colistin resistance was observed in *Klebsiella pneumoniae* isolates from bloodstream infections in the year 2016 to 2019, while a 12.90% (4/31) pooled colistin resistance was observed in *Klebsiella pneumoniae* isolates from bloodstream infections in the year 2020 and after.

### 3.3. Antibiotic Resistance Rate in the Studied Klebsiella pneumoniae 

Since there was high heterogeneity (I^2^ > 50%) in the prevalence of antibiotic resistance of *Klebsiella pnuemoniae* in the study of the 5 studies that had multi-drug resistant (MDR), a pooled MDR of 80.1%, 95% CI (65.0–95.2%) was observed, amikacin showed a 25.4%, 95% CI (8.0–42.7%) pooled resistance, ciprofloxacin showed a 45.3%, 95% CI (13.2–77.5%) pooled resistance from five studies, ertapenem, imipenem and meropenem showed a 44.4%, 95% CI (19.9–68.8%), 35.2%, 95% CI (18.9–51.5%) and 36.1%, 95% CI (19.6–52.5%) pooled resistance, respectively, while gentamicin and tigecycline showed 33.3%, 95% CI (7.4–59.2%) and 5.1%, 95% CI (0.3–10.0%) pool resistance in the studied *Klebsiella pneumoniae* from bloodstream infections as illustrated in Table 1.

### 3.4. Pooled Colistin Resistance Rate in the Studied Klebsiella pneumoniae from Bloodstream Infections by Country of Study and Isolates Sources

As shown in Table 2, *Klebsiella pneumoniae* bloodstream infection from Thailand showed the highest rate of resistance 19.2%, 95% CI (4.1–34.4%), Brazil followed this with a reported colistin resistance rate of 14.6%, 95% CI (4.6–24.6%). Isolates from Pakistan showed 12.9%, 95% CI (1.1–24.7%) colistin resistance, a 2.8%, 95% CI (1.4–6.1%), and 7.6%, 95% CI (3.9–19.0%) pooled colistin resistance was observed in bloodstream infection caused by *Klebsiella pneumoniae* from China and India, respectively. However, South Korea reported the least colistin resistance, 0.8%, 95% CI (0.2–1.6%) in *Klebsiella pneumoniae* isolates from bloodstream infections among countries that have reported colistin resistance using the established method.

Figure 5 shows the colistin resistance in *Klebsiella pneumoniae* from bloodstream infection by isolated sources. The *Klebsiella pneumoniae* recovered from the intensive care unit (ICU) showed a colistin resistance of 11.5% (9/78). Other sources showed 3.03% (80/2604) resistance. 

## 4. Discussion

Multidrug-resistant gram-negative bacteria (MDR) threaten public health globally [33]. Among gram-negative bacteria, *Klebsiella pneumoniae* is one of the most common causes of healthcare-associated infections (HALs) [34]. *Klebsiella pneumoniae* associated with healthcare is highly resistant to several antibiotics, especially carbapenems [35,36]. In the fight against carbapenem-resistant *Klebsiella pneumoniae*, colistin (polymyxin E) is now commonly used [37]. In the intensive care unit (ICU), patients with bloodstream infections caused by carbapenem-resistant *Klebsiella pneumoniae* have poorer outcomes, resulting in a rising mortality rate globally [37,38]. Therefore, it is essential to understand the antimicrobial resistance profile of *Klebsiella pneumoniae* from bloodstream infection to provide interventions, including those for infection prevention and control. The data from 2016 through 2022 were included in this comprehensive review and meta-analysis on colistin and other antibiotic resistance in *Klebsiella pneumoniae* from bloodstream infections.

Colistin usage was popular in the 1940s until it was found to be very harmful to the kidneys and the nervous system in the 1970s, which led to its removal from use as a systemic antibacterial treatment [39]. At the same time, drugs with less harmful side effects were readily accessible and efficient, but this was short-lived with the development of resistance to those new antibiotics. As a result of the proliferation of carbapenem-resistant Gram-negative bacteria, polymyxins have become increasingly used as a medicine of last resort [40]. Globally, the use of colistin almost quadrupled from 2006 to 2012 [39,40]. Our study recorded a pooled colistin resistance prevalence of 3.1% in *Klebsiella pneumoniae* from bloodstream infections. Our findings contradicted a 10% colistin resistance in *Klebsiella pneumoniae* found in a recent clinical study involving clinical samples other than blood culture [41]. Our review was focused on bloodstream-infected *Klebsiella pneumoniae* resistant to colistin, with only ten studies using the recommended method to test antibiotic susceptibility testing. There is a need to develop a surveillance system that will include quality laboratory processes using the CLSI and EUCAST recommended microdilution method for antibiotic susceptibility testing for colistin resistance. 

The observation from this study showed that there is a need to advocate the use of the right method in clinical settings to ensure the right insight into the actual colistin resistance rate globally. Although several studies have been conducted on colistin resistance, as shown in a recent bibliometric study of published articles on colistin resistance by *Klebsiella pneumoniae,* the actual resistance rate was not determined due to the heterogeneity in methods [34]. 

The high colistin resistance observed in bloodstream *Klebsiella pnuemoniae* isolates studied from 2020 (the COVID-19 pandemic years) compared to the pre-COVID-19 period could be attributed to possible high colistin usage in clinical settings during the pandemic to treat carbapenem-resistant *Klebsiella pneumoniae*. This hypothesis is worth investigating to determine if the COVID-19 pandemic caused an increased colistin administration in hospital settings. To support our assertion, a recent study [40] which reported a case series of carbapenem-resistant *Klebsiella pneumoniae* infection with COVID-19 and how it was managed, showed that colistin was used in the two cases with one survivor. Arteaga-Livias et al. [40] explored how colistin became an indispensable drug in patients with carbapenem-resistant *Klebsiella pneumoniae* and COVID-19 co-infections. 

Due to our study’s high heterogeneity (Figure 2 and Figure 3), the outcome cannot be generalized to the population; therefore, caution should be applied when interpreting the study result. We used the random effect model due to dissimilarities in the studies used to account for this concern. 

Bloodstream infection is particularly of concern as this can result in high mortality rates, especially those caused by carbapenem and colistin resistance. Understanding colistin resistance is important because there are no newer drugs in the pipelines. As demonstrated by our study, the average multidrug-resistant (MDR) *Klebsiella pneumoniae* from bloodstream infection was over 70%. Moderate amikacin resistance was observed in the study. Amikacin is a drug of choice in treating multi-drug-resistant *Klebsiella pneumoniae*. The increased resistance in isolates that are not even carbapenem-resistant is a concern as this has caused increased use of carbapenems, making them develop resistance quickly after exposure. This resistance showed that antibiotic therapy options available for the clinician are becoming narrower as time passes. 

Bloodstream infections due to *Klebsiella pneumoniae* from the intensive care unit were more colistin-resistant than non-ICU. This finding agrees with the recent study in the United States, where resistance rates of bacterial isolates from ICU and non-ICU patients were compared. Higher resistance rate was reported for pathogens from ICU patients [41]. High colistin resistance in the ICU could be attributed to the increased use of colistin in ICU patients, especially as high carbapenem-resistant *Klebsiella pneumoniae* has been reported in the ICU by several studies [41,42,43]. Most carbapenem-resistant *Klebsiella pneumoniae* are resistant to the most available antibiotics; as a result, colistin is usually used in some cases. It has been reported that some experimental drugs have been designed and used, while some are at different phases of a clinical trial [44,45,46,47,48].

Although the data reported in this meta-analysis is not distributed in all continents, this meta-analysis reemphasizes the reemergence of colistin resistance due to the increased use of colistin for therapeutic purposes in carbapenem resistance in *Klebsiella pneumoniae*. Moreover, this meta-analysis provides more robust results that can help researchers and policy makers understand the magnitudes of the problem of increased colistin resistance as seen in data from 2020 and beyond giving insights to design an actionable solution.

## 5. Limitation of the Study

The studies included in this study were highly heterogeneous, being from different geographical locations, social-economic differences and also the difference in health care systems, including antimicrobial resistance control policy. These, therefore, hamper the process of generalizing the outcome to the general population. There was insufficient data for correlational analysis as most studies reported epidemiological data with limited clinical outcomes information. Furthermore, data for analysis in some aspects of the study were inadequate. This inadequate data may have therefore impacted the actual possible situation in the general population. Most countries had no reports, as such distribution across continents was not done. This inadequate of distribution of colistin resistance data across continents was mostly affected by wrong AST methods being used by most countries. 

## 6. Conclusions

This study is the first systematic review and meta-analysis that examines *Klebsiella pneumoniae* colistin resistance from bloodstream infection to the best of our knowledge. This study showed low colistin resistance in *Klebsiella pneumoniae* isolates from bloodstream infections. However, significant colistin resistance was observed in isolates collected from 2020 and beyond. Significant colistin resistance was also observed in *Klebsiella pneumoniae* isolates in bloodstream infections from ICU compared to those from non-ICUs. As a result, there is a need to institute colistin administration stewardship in the ICU in clinical settings. There is a need for proper surveillance of colistin resistance using the right methods to give better insight into it and better prepare policymakers and the scientific community to manage increased resistance. Due to colistin toxicity, using the right techniques for its susceptibility will ensure patients are not unduly exposed due to wrong AST. There are guidelines for correctly testing the sensitivity of bacteria to colistin. They must be respected; otherwise, the results of the AST must be disregarded.

## Figures and Tables

**Figure 1 pathogens-11-01092-f001:**
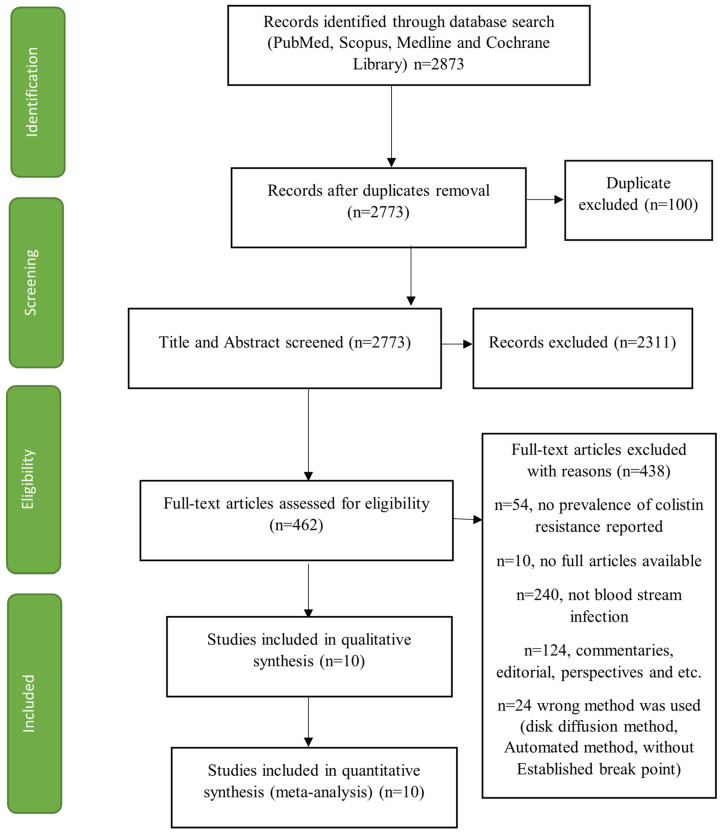
Flow chart of the literature screening process.

**Figure 2 pathogens-11-01092-f002:**
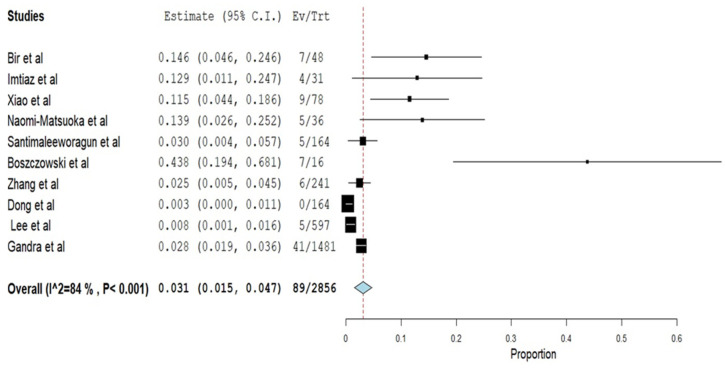
Forest Plot of Prevalence of Colistin Resistance of included studies.

**Figure 3 pathogens-11-01092-f003:**
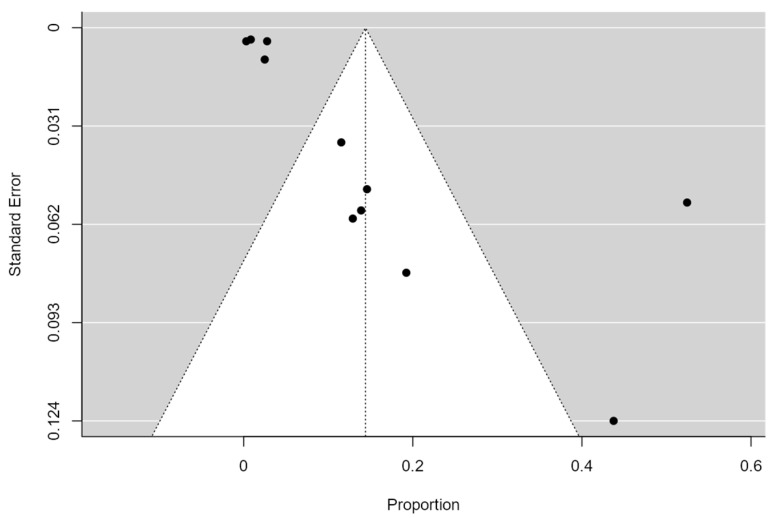
A funnel plot based on the Colistin Resistance of included studies evaluated publication bias.

**Figure 4 pathogens-11-01092-f004:**
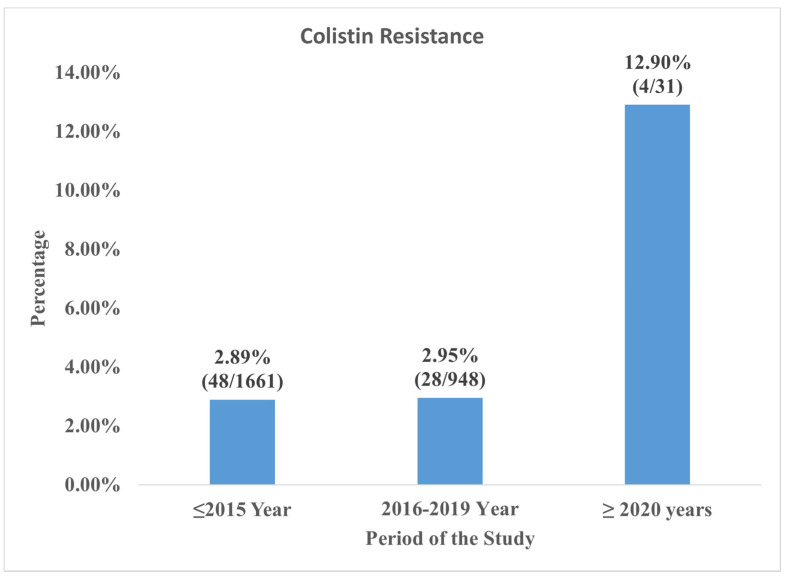
Time Trend Meta-analysis of Colistin Resistance in *Klebsiella pneumoniae* from Blood Stream Infection by Year of Study.

**Figure 5 pathogens-11-01092-f005:**
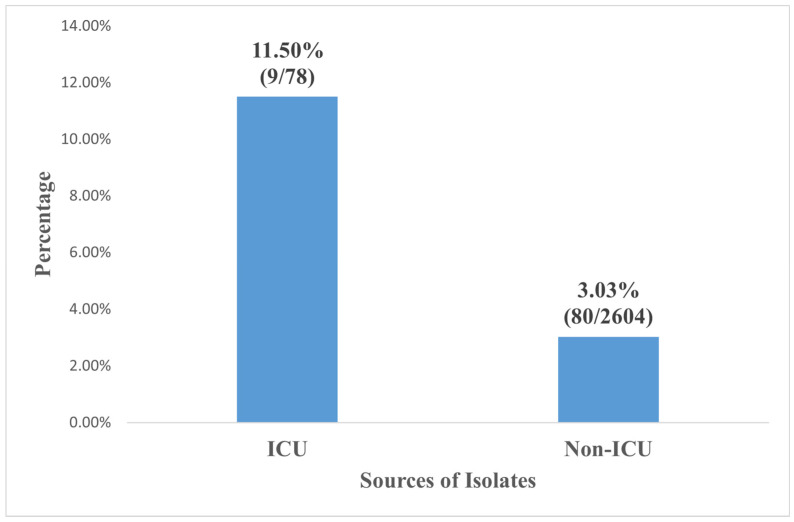
Colistin Resistance in *Klebsiella pneumoniae* from Blood Stream Infection by Isolates Source.

**Table 1 pathogens-11-01092-t001:** Pooled Antibiotic Resistance Rate in *Klebsiella pneumoniae* isolates from bloodstream infections.

Antibiotic	Number of Studies	n/N	Resistance (%)	95%CI	
				Lower Limit	Upper Limit
MDR	5	235/357	80.1	65.0	95.2
Amikacin	4	121/1080	25.4	8.0	42.7
Ciprofloxacin	5	1372/1980	45.3	13.2	77.5
Ertapenem	3	787/1800	44.4	19.9	68.8
Gentamicin	4	223/1080	33.3	7.4	59.2
Imipenem	5	818/1980	35.2	18.9	51.5
Meropenem	6	838/2144	36.1	19.6	52.5
Tigercycline	3	25/483	5.1	0.3	10.0

Keys: n = number of outcome, N = Total of number tested, CI = confidence interval, % = Percent.

**Table 2 pathogens-11-01092-t002:** Prevalence of Colistin Resistance in *Klebsiella pneumoniae* isolates from bloodstream infections by Country Study.

Country	Study Period (year/s)	Number of Study	n/N	Resistance (%)	95%CI	
					Lower Limit	Upper Limit
Brazil	2010–2015	1	7/48	14.6	4.6	24.6
China	2011–2019	3	15/483	2.8	1.4	6.1
India	2008–2019	2	48/1529	7.6	3.9	19.0
Pakistan	2021	1	4/31	12.9	1.1	24.7
Peru	2018	1	5/36	13.9	2.6	15.2
South Korea	2016–2017	1	5/597	0.8	0.2	1.6
Thailand	2017–2018	1	5/26	19.2	4.1	34.4

Keys: n = number of outcomes, N = Total of number tested, CI = confidence interval, % = Percent.

## Data Availability

The data used for this study is attached as Appendix A.

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
