# Peer review of "Global Prevalence of Colistin Resistance in Klebsiella pneumoniae from Bloodstream Infection: A Systematic Review and Meta-Analysis"

_pathogens, 2022, doi:10.3390/pathogens11101092_

Round 1
Reviewer 1 Report
Authors have done meta-analysis to recover the available data on prevalence.
Authors are requested the following queries:
1) The study which were included in the study are of different geo-graphical location, socio-economic difference and different health care system and, as different AST prectice(as mentioned). Authors are requested to explain this point certainly to the manuscript as a limitation or critical gap or as an perspective with additionally what steps at national or international level can be taken to restrict the resistance!
2) Can you explain how will you justifiy the authenticity of the data and prevalence ? as it is meta-study, you can have proper justification in to manuscript that data are of primary , initial and give certain initial warning for the resistance.
3) As authors mentions regarding AST practice and making generalized statement, which could not inferred the conclusion which you have draw, but this study is still important for the healt care practice, so you are requested to add additional paragraph at the end of disucssion regards to the importance of having meta analysis based data!
Author Response
General Response: Thank you very much for taking the time to review our manuscript, which has improved the quality. We have also edited the entire manuscript for grammar and spelling errors.
Comment 1: The study which were included in the study are of different geo-graphical location, socio-economic difference and different health care system and, as different AST prectice(as mentioned). Authors are requested to explain this point certainly to the manuscript as a limitation or critical gap or as an perspective with additionally what steps at national or international level can be taken to restrict the resistance!
Author response: Thank you for your comment. We have included your suggestion in the limitation aspect of the study. Kindly see lines 326 to 328. Also, the other aspect of the study limitation and conclusion speaks to what needs to be done nationally and internationaly regarding combating colistin resistance.
Comment 2: Can you explain how will you justifiy the authenticity of the data and prevalence ? as it is meta-study, you can have proper justification in to manuscript that data are of primary , initial and give certain initial warning for the resistance.
Author response: Thank you for this comment. We have modified the inclusion criteria, which addressed your concern. Kindly see lines 137 to 145.
Comment 3: As authors mentions regarding AST practice and making generalized statement, which could not inferred the conclusion which you have draw, but this study is still important for the healt care practice, so you are requested to add additional paragraph at the end of disucssion regards to the importance of having meta analysis based data!
Author response: Thank you for your suggestion. We have added the suggested paragraph. Kindly see lines 319 to 325.
Reviewer 2 Report
Generally well written,
Scope for being somewhat more concise.
For example, in the abstract “There are guidelines for correctly testing 74 the sensitivity of bacteria to colistin. They must be respected; otherwise, the results of the AST must 75 be disregarded.” Is not directly relevant to your study and could be deleted. Also lines 275 to 279 are irrelevant
Also the back ground describes many studies that might appear in the meta-analysis.
Can you be a bit more clear on inclusion and exclusion criteria? Presumably these bacteremia cases are all hospital inpatients? Paediatric age groups?
It is unfortunate if you exclude 24 studies with the wrong method [leaving only 10 studies in your sample]. In those 24 studies it would be of interest to see if there were nay systematic difference [or similarities].
The methods need to state which method you used to calculate the 95% confidence intervals [there are several methods]
Also, I suspect that there are more than 10 eligible studies out there.
Have a look in the literature searches I have done
Hurley JC. Candida–Acinetobacter–Pseudomonas Interaction Modelled within 286 ICU Infection Prevention Studies. Journal of Fungi. 2020;6(4):252.
My personal preference is to use a random-effect model to analyse data regardless of amounts of heterogeneity [especially noting what you state at line 290].
The scale on figure 2 is linear this would be better if it was logistic [or log] transformed, given the number of results under 0.1 %.
Author Response
General Response: Thank you very much for taking the time to review our manuscript, which has improved the quality. We have also edited the entire manuscript for grammar and spelling errors.
Comment 1: For example, in the abstract, “There are guidelines for correctly testing 74 the sensitivity of bacteria to colistin. They must be respected; otherwise, the results of the AST must 75 be disregarded.” Is not directly relevant to your study and could be deleted. Also lines 275 to 279 are irrelevant.
Author response: Thank you for your suggestions. We have deleted those statements in those sections; kindly see the marked sections on lines 74 to 76 and 280 to 284.
Comment 2: Also the back ground describes many studies that might appear in the meta-analysis.
Author response: Thank you for your assertion. We had re-examined those studies in the background they did not meet the inclusion criteria. They are neither from bloodstream infection nor used methods other than the microdilution method, which is the only recommended method by EUCAST and CLSI. It is only through the micro-dilution method that a breakpoint has been established. We have outlined those criteria in the inclusion ad exclusion section of the manuscript. Examples of some of those articles you spoke about are;
- Azam M, Gaind R, Yadav G, Sharma A, Upmanyu K, Jain M, et al. Colistin Resistance Among Multiple Sequence Types of Klebsiella pneumoniae Is Associated With Diverse Resistance Mechanisms: A Report From India. Front Microbiol. 2021;12 February- REASON FOR EXCLUSION- The method used was not Micro dilution.
- Monaco M, Giani T, Raffone M, Arena F, Garcia-Fernandez A, Pollini S, et al. Colistin resistance superimposed to endemic carbapenem-resistant Klebsiella pneumoniae: A rapidly evolving problem in Italy, November 2013 to April 2014. Eurosurveillance. 2014;19:14–8. REASON FOR EXCLSUION- The result for colistin resistance for Klebsiella pneumoniae from bloodstream infection was not reported independently of those from other sample sources.
Comment 3: Can you be a bit more clear on inclusion and exclusion criteria? Presumably these bacteremia cases are all hospital inpatients? Paediatric age groups?
Author response: Thank you, all the studies were hospital-based; there were no specifics in most of the studies regarding if they were inpatients or outpatients. Also, the data were from different age groups. We know this as mean age was reported in most studies; we had reported this as a limitation that hampers correlational studies in the study limitation section. Kindly see lines 330 to 334.
Comment 4: It is unfortunate if you exclude 24 studies with the wrong method [leaving only 10 studies in your sample]. In those 24 studies it would be of interest to see if there were nay systematic difference [or similarities].
Author response: It was really unfortunate that we had to exclude 24 studies due to wrong methods. Your suggestion regarding checking for similarities or differences is a perfect idea. However, we did not design this study to check those; doing that will mean we will have to design the study; however, we had suggested this in the limitation as a possible study that could be undertaken in the future.
Comment 5: The methods need to state which method you used to calculate the 95% confidence intervals [there are several methods]
Author response: Thank you for the observation. We used the Jackson method of the DelSimonian-Laird (DL) model in OpenMeta-Analyst. We have included it in the method section; kindly see lines 172 to 176.
Comment 6: Also, I suspect that there are more than 10 eligible studies out there.
Author response: We understand your concerns. However, we did another search, and the outcomes were the same; search strategies used to the best of our knowledge will limit missing out on any eligible study within the inclusion criteria set.
Comment 7: Have a look in the literature searches I have done, Hurley JC. Candida–Acinetobacter–Pseudomonas Interaction Modelled within 286 ICU Infection Prevention Studies. Journal of Fungi. 2020;6(4):252.
Author response: Thank you for the kind help and suggestion of this article, we looked at it and learned from it, and we used it to refine our inclusion criteria which you can in the method section on lines 137 to 145.
Comment 8: My personal preference is to use a random-effect model to analyse data regardless of amounts of heterogeneity [especially noting what you state at line 290].
Author response: Thank you for pointing us to this idea, we used the random effect model in the entire analysis, and we initially tried to give context to why we used the random effect model; we had written it in a better form; see lines 170 to 176.
Comment 9: The scale on figure 2 is linear this would be better if it was logistic [or log] transformed, given the number of results under 0.1 %.
Author response: Thank you for your suggestion. We had preferred it in linear for easy reference since when transformed, and interpretation will no longer be direct.